# Pleiotropic mutations can rapidly evolve to directly benefit self and cooperative partner despite unfavorable conditions

Samuel Frederick Mock Hart[1], Chi-Chun Chen[1], Wenying Shou[1,2]*

[1]Fred Hutchinson Cancer Research Center, Division of Basic Sciences, Seattle, United States; [2]University College London, Department of Genetics, Evolution and Environment, Centre for Life's Origins and Evolution (CLOE), London, United Kingdom

**Abstract** Cooperation, paying a cost to benefit others, is widespread. Cooperation can be promoted by pleiotropic 'win-win' mutations which directly benefit self (self-serving) and partner (partner-serving). Previously, we showed that partner-serving should be defined as increased benefit supply rate per intake benefit. Here, we report that win-win mutations can rapidly evolve even under conditions unfavorable for cooperation. Specifically, in a well-mixed environment we evolved engineered yeast cooperative communities where two strains exchanged costly metabolites, lysine and hypoxanthine. Among cells that consumed lysine and released hypoxanthine, *ecm21* mutations repeatedly arose. *ecm21* is self-serving, improving self's growth rate in limiting lysine. *ecm21* is also partner-serving, increasing hypoxanthine release rate per lysine consumption and the steady state growth rate of partner and of community. *ecm21* also arose in monocultures evolving in lysine-limited chemostats. Thus, even without any history of cooperation or pressure to maintain cooperation, pleiotropic win-win mutations may readily evolve to promote cooperation.

*For correspondence:
wenying.shou@gmail.com

Competing interests: The authors declare that no competing interests exist.

## Introduction

Cooperation, paying a fitness cost to generate benefits available to others, is widespread and thought to drive major evolutionary transitions (*Maynard Smith, 1982*; *Sachs et al., 2004*). For example, in multicellular organisms, different cells must cooperate with each other and refrain from dividing in a cancerous fashion to ensure the propagation of the germline (*Michod and Roze, 2001*). Cooperation between species, or mutualistic cooperation, is also common (*Boucher, 1985*). In extreme cases, mutualistic cooperation is obligatory, that is, cooperating partners depend on each other for survival (*Cook and Rasplus, 2003*; *Zientz et al., 2004*). For example, insects and endosymbiotic bacteria exchange costly essential metabolites (*Zientz et al., 2004*; *Gil and Latorre, 2019*).

Cooperation is vulnerable to 'cheaters' who gain a fitness advantage over cooperators by consuming benefits without reciprocating fairly. Cancers are cheaters of multicellular organisms (*Athena et al., 2015*), and rhizobia variants can cheat on their legume hosts (*Gano-Cohen et al., 2019*). How might cooperation survive cheaters?

Various mechanisms are known to protect cooperation against cheaters. In 'partner choice,' an individual preferentially interacts with cooperating partners over spatially equivalent cheating partners (*Sachs et al., 2004*; *Kiers et al., 2003*; *Bshary and Grutter, 2006*; *Shou, 2015*). For example, client fish observes cleaner fish cleaning other clients and then chooses the cleaner fish that offers high-quality service (removing client parasites instead of client tissue) to interact with (*Bshary and Grutter, 2006*).

For organisms lacking partner choice mechanisms, a spatially structured environment can promote the origin and maintenance of cooperation (*Sachs et al., 2004*; *Chao and Levin, 1981*; *Momeni et al., 2013a*; *Harcombe, 2010*; *Nowak, 2006*; *Pande et al., 2016*; *Harcombe et al., 2018*). This is because in a spatially structured environment, neighbors repeatedly interact, and thus cheaters will eventually suffer as their neighbors perish (partner fidelity feedback). In a well-mixed environment, since all individuals share equal access to the cooperative benefit regardless of their contributions, cheaters are favored over cooperators (*Harcombe, 2010*). An exception is that cooperators can stochastically purge cheaters if cooperators happen to be better adapted to an environmental stress than cheaters (*Asfahl et al., 2015*; *Morgan et al., 2012*; *Waite and Shou, 2012*). Finally, pleiotropy – a single mutation affecting multiple phenotypes – can stabilize cooperation if reducing benefit supply to partner also elicits a crippling effect on self (*Foster et al., 2004*; *Dandekar et al., 2012*; *Oslizlo et al., 2014*; *Harrison and Buckling, 2009*; *Sathe et al., 2019*). For example, when the social amoeba *Dictyostelium discoideum* experience starvation and form a fruiting body, a fraction of the cells differentiates into a non-viable stalk in order to support the remaining cells to differentiation into viable spores. *dimA* mutants attempt to cheat by avoiding the stalk fate, but they also fail to form spores (*Foster et al., 2004*). In this case, a gene links an individual's partner-serving trait to its self-serving trait, thus stabilizing cooperation.

To date, pleiotropic linkage between a self-serving trait and a partner-serving trait has been exclusively demonstrated in systems with long evolutionary histories of cooperation. Thus, it is unclear how easily such a genetic linkage can arise. One possibility is that cooperation promotes pleiotropy. Indeed, theoretical work suggests that cooperation can stabilize pleiotropy (*Dos Santos et al., 2018*), and that as cooperators evolve to resist cheater invasion, pleiotropic linkage between self-serving and partner-serving traits is favored (*Frénoy et al., 2013*). A second possibility is that pleiotropy promotes cooperation (*Foster et al., 2004*; *Dandekar et al., 2012*; *Oslizlo et al., 2014*; *Harrison and Buckling, 2009*; *Sathe et al., 2019*). These two possibilities are not mutually exclusive.

Here, we investigate whether pleiotropic 'win-win' mutations directly benefiting self and directly benefiting partner could arise and stabilize nascent cooperation. We test this in a synthetic cooperative community growing in an environment unfavorable for cooperation (e.g. in a well-mixed environment or in monocultures without the cooperative partner). The community is termed CoSMO (Cooperation that is Synthetic and Mutually Obligatory). CoSMO comprises two non-mating engineered *Saccharomyces cerevisiae* strains: $L^-H^+$ requires lysine ($L$) and pays a fitness cost to overproduce hypoxanthine ($H$, an adenine derivative) (*Waite and Shou, 2012*; *Hart et al., 2019a*), while $H^-L^+$ requires hypoxanthine and pays a fitness cost to overproduce lysine (*Hart et al., 2019b*; *Figure 1A*). Overproduced metabolites are released into the environment by live cells (*Hart et al., 2019a*), allowing the two strains to feed each other. CoSMO models the metabolic cooperation between certain gut microbial species (*Rakoff-Nahoum et al., 2016*) and between legumes and rhizobia (*Schubert, 1986*), as well as other mutualisms (*Beliaev et al., 2014*; *Helliwell et al., 2011*; *Carini et al., 2014*; *Rodionova et al., 2015*; *Zengler and Zaramela, 2018*; *Jiang et al., 2018*). Similar to natural systems, in CoSMO exchanged metabolites are costly to produce (*Waite and Shou, 2012*; *Hart et al., 2019a*), and cooperation can transition to competition when the exchanged metabolites are externally supplied (*Momeni et al., 2013b*). Importantly, principles learned from CoSMO have been found to operate in communities of un-engineered microbes. These include how fitness effects of interactions might affect spatial patterning and species composition in two-species communities (*Momeni et al., 2013b*), as well as how cooperators might survive cheaters (*Momeni et al., 2013a*; *Waite and Shou, 2012*; see Discussions in these articles).

In our previous work, we allowed nine independent lines of CoSMO to evolve for over 100 generations in a well-mixed environment by performing periodic dilutions (*Hart et al., 2019a*; *Shou et al., 2007*). Throughout evolution, the two cooperating strains coexisted due to their metabolic codependence (*Momeni et al., 2013b*; *Shou et al., 2007*). In a well-mixed environment, since partner-supplied benefits are uniformly distributed and equally available to all individuals, a self-serving mutation will be favored regardless of how it affects the partner. Indeed, all characterized mutants isolated from CoSMO displayed self-serving phenotypic changes (e.g. improved affinity for partner-supplied metabolites; *Waite and Shou, 2012*; *Hart et al., 2019a*; *Hart et al., 2019b*), outcompeting

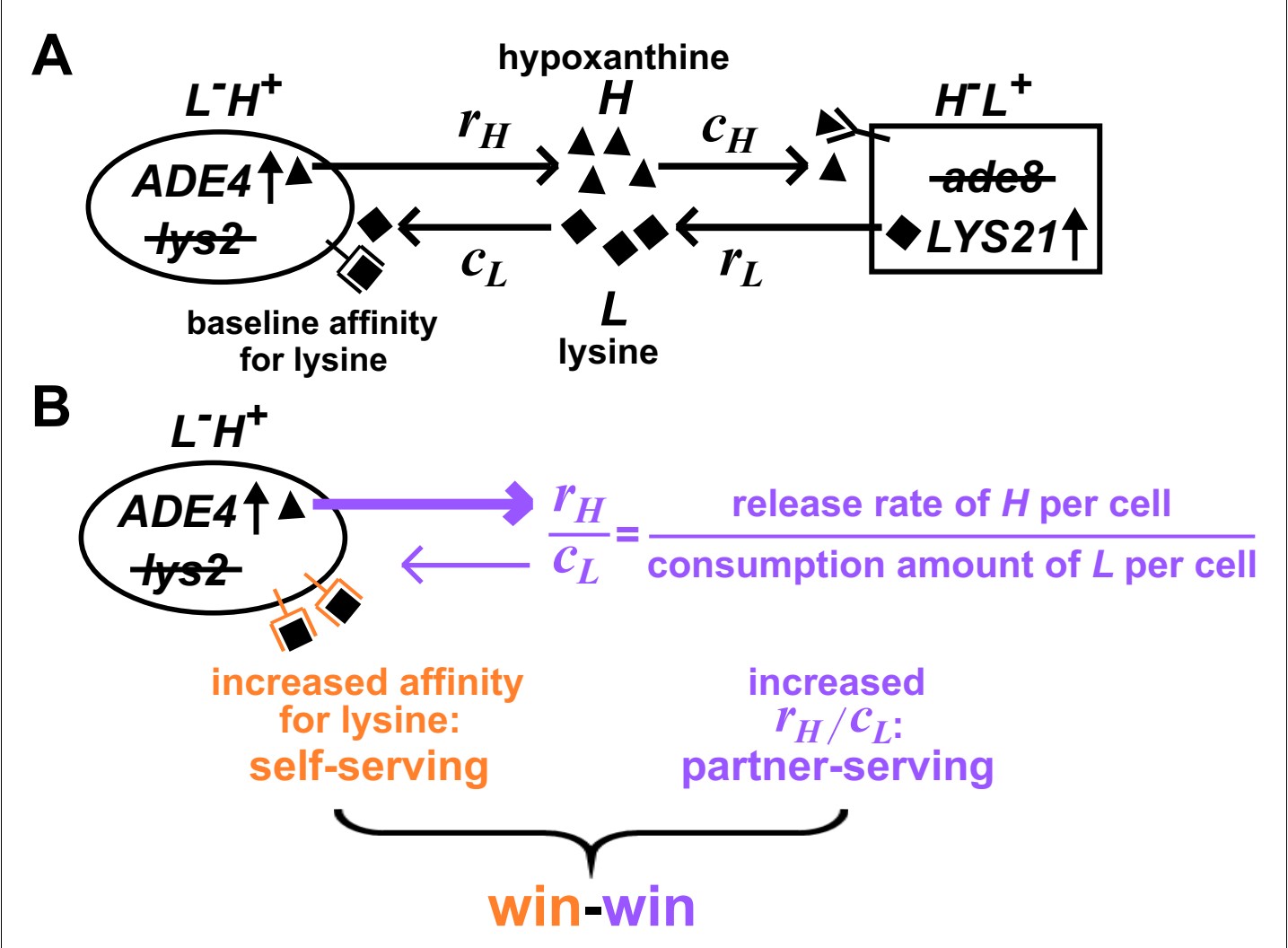

**Figure 1.** Win-win mutation in a nascent cooperative community. (**A**) CoSMO (Cooperation that is Synthetic and Mutually Obligatory) consists of two non-mating cross-feeding yeast strains, each engineered to overproduce a metabolite required by the partner strain. Metabolite overproduction is due to a mutation that renders the first enzyme of the biosynthetic pathway resistant to end-product inhibition (*Armitt and Woods, 1970*; *Feller et al., 1999*). Hypoxanthine and lysine are released by live $L^-H^+$ and live $H^-L^+$ cells at a per cell rate of $r_H$ and $r_L$, respectively (*Hart et al., 2019a*), and are consumed by the partner at a per cell amount of $c_H$ and $c_L$, respectively. The two strains can be distinguished by different fluorescent markers. (**B**) Win-win mutation. A pleiotropic win-win mutation confers a self-serving phenotype (orange) and a partner-serving phenotype (lavender).

their ancestor in community-like environments. Here, we report the identification of a pleiotropic win-win mutation which is both self-serving and partner-serving. This win-win mutation also arose in the absence of the cooperative partner. Thus, cooperation-promoting win-win mutations can arise in a community without any evolutionary history of cooperation and in environments unfavorable to cooperation. Our work suggests the possibility that pre-existing pleiotropy can stabilize nascent cooperation in natural communities.

## Results

### Criteria of a win-win mutation

A win-win mutation is defined as a single mutation (e.g., a point mutation, a translocation, a chromosome duplication) that directly promotes the fitness of self (self-serving) and the fitness of partner (partner-serving). To define 'direct' here, we adapt the framework from Chapter 10 of *Peters et al.,*

*2017*: A mutation in genotype *A* exerts a direct fitness effect on genotype *B* if the mutation can alter the growth rate of *B* even if the biomass of *A* is fixed (*Hart et al., 2019b*).

For $L^-H^+$, a self-serving mutation should improve the growth rate of self by, for example, increasing cell's affinity for lysine (*Figure 1B*, orange). A self-serving mutation allows the mutant to outcompete a non-mutant. A partner-serving mutation should improve the growth rate of partner at a fixed self biomass. Since the partner requires hypoxanthine, a partner-serving mutation in $L^-H^+$ should increase the hypoxanthine supply rate per $L^-H^+$ biomass. Since the biomass of $L^-H^+$ is linked to lysine consumption, the partner-serving phenotype of $L^-H^+$ translates to hypoxanthine supply rate per lysine consumption, or equivalently, hypoxanthine release rate per cell ($r_H$) normalized by the amount of lysine consumed to make a cell ($c_L$) (*Hart et al., 2019b*). We call this ratio $r_H/c_L$ 'H-L exchange ratio' (*Figure 1B*, purple), which can be interpreted as the yield coefficient while converting lysine consumption to hypoxanthine release. Note that a partner-serving mutation will eventually feedback to promote self-growth. Indeed, after an initial lag, the growth rate of partner, of self, and of the entire community reach the same steady state growth rate $\sqrt{\frac{r_H}{c_L}\frac{r_L}{c_H}}$, where $r_L$ (lysine release rate per cell) and $c_H$ (hypoxanthine consumption amount per cell) are phenotypes of $H^-L^+$ (*Hart et al., 2019b*).

## Community and monoculture evolution share similar mutations

We randomly isolated evolved $L^-H^+$ colonies from CoSMO and subjected them to whole-genome sequencing. Nearly every sequenced clone harbored one or more of the following mutations: *ecm21*, *rsp5*, and duplication of chromosome 14 (*DISOMY14*) (*Table 1*, top), consistent with our earlier studies (*Waite and Shou, 2012*; *Hart et al., 2019a*; *Hart et al., 2019b*; *Green et al., 2020*). Mutations in *RSP5*, an essential gene, mostly involved point mutations (e.g., *rsp5(P772L)*), while mutations in *ECM21* mostly involved premature stop codons and frameshift mutations (*Table 1*, top; *Figure 2—figure supplement 1*). Similar mutations also repeatedly arose when $L^-H^+$ evolved as a monoculture in lysine-limited chemostats (*Table 1*, bottom), suggesting that these mutations emerged independently of the partner.

## Self-serving mutations increase the abundance of metabolite permease on cell surface

Evolved $L^-H^+$ clones are known to display a self-serving phenotype: they could form microcolonies on low-lysine plates where the ancestor failed to grow (*Waite and Shou, 2012*; *Hart et al., 2019b*). To quantify this self-serving phenotype, we used a fluorescence microscopy assay (*Hart et al., 2019c*) to measure the growth rates of ancestral and evolved $L^-H^+$ in various concentrations of lysine. Under lysine limitation characteristic of the CoSMO environment (*Figure 2A*, 'Comm. environ.'), evolved $L^-H^+$ clones containing an *ecm21* or *rsp5* mutation grew faster than a *DISOMY14* strain which, as we showed previously, grew faster than the ancestor (*Hart et al., 2019b*). An engineered *ecm21Δ* or *rsp5(P772L)* mutation was sufficient to confer the self-serving phenotype (*Figure 2A*). In competition experiments in lysine-limited chemostats (8 hr doubling time), *ecm21Δ* rapidly outcompeted ancestral cells since *ecm21Δ* grew 4.4 times as fast as the ancestor (*Figure 2—figure supplement 2*).

A parsimonious explanation for *ecm21*'s self-serving phenotype is that during lysine limitation, the high-affinity lysine permease Lyp1 is stabilized on cell surface in the mutant. We have previously shown that duplication of the *LYP1* gene, which resides on chromosome 14, is necessary and sufficient for the self-serving phenotype of *DISOMY14* (*Hart et al., 2019b*). Rsp5, an E3 ubiquitin ligase, is recruited by various 'adaptor' proteins to ubiquitinate and target membrane transporters, including Lyp1, for endocytosis and vacuolar degradation (*Lin et al., 2008*). In high lysine, Lyp1-GFP was localized to both cell membrane and vacuole in ancestral and *ecm21* cells, but localized to the cell membrane in *rsp5* cells (*Figure 2B*, top row). Thus, Lyp1 localization was normal in *ecm21* but not in *rsp5*, consistent with the notion that at high lysine concentrations, Lyp1 is targeted for ubiquitination by Rsp5 through the Art1 instead of the Ecm21 adaptor (*Lin et al., 2008*). When ancestral $L^-H^+$ was incubated in low lysine, Lyp1-GFP was initially localized on the cell membrane to facilitate lysine uptake, but later targeted to the vacuole for degradation and recycling (*Jones et al., 2012*; *Figure 2B*, left column middle and bottom panels). However, in both *ecm21Δ* and *rsp5(P772L)* mutants, Lyp1-GFP was stabilized on cell membrane during prolonged lysine limitation (*Figure 2B*,

**Table 1.** Mutations that repeatedly arose in independent lines.

Single-nucleotide polymorphisms (SNPs) and chromosomal duplications from Illumina re-sequencing of $L^-H^+$ from <u>Co</u>operation that is <u>S</u>ynthetic and <u>M</u>utually <u>O</u>bligatory (CoSMO) communities (top) and lysine-limited chemostats (bottom). All clones except for two (WY1592 and WY1593 of line B3 at Generation 14) had either an *ecm21* or an *rsp5* mutation, often in conjunction with chromosome 14 duplication. Note that the RM11 strain background in this study differed from the S288C strain background used in our earlier study (**Waite and Shou, 2012**). This could explain, for example, why mutations in *DOA4* were repeatedly observed in the earlier study (**Waite and Shou, 2012**) but not here. For a schematic diagram of the locations of mutations with respect to protein functional domains in *ecm21* and *rsp5*, see **Figure 2—figure supplement 1**. For other mutations, see **Table 1—source data 1**.

| $L^-H+$ | Line | Generation | *ecm21* | *rsp5* | Chromosome duplicated | Strain |
|---|---|---|---|---|---|---|
| CoSMO comm. | A1 | 24 | Glu316 -> Stop | – | 11, 14 | WY1588 |
| | | | – | Pro772 -> Leu | 11 | WY1589 |
| | | 151 | – | Pro772 -> Leu | – | WY1590 |
| | | | – | Pro772 -> Leu | 11, 14, 16 | WY1591 |
| | B1 | 25 | Leu812->Stop | – | 14 | WY1584 |
| | | 49 | – | Gly689 -> Cys | 14 | WY1585 |
| | | 76 | – | Gly689 -> Cys | – | WY1586 |
| | | | – | Gly689 -> Cys | 14 | WY2467 |
| | | | – | Gly689 -> Cys | 14 | WY1587 |
| | B3 | 14 | – | – | – | WY1592 |
| | | | – | – | 14 | WY1593 |
| | | 34 | Arg742 -> Stop | – | 14, 16 | WY1594 |
| | | | Arg742 -> Stop | – | 14, 16 | WY1595 |
| | | 63 | Arg742 -> Stop | Arg742 -> His | 12, 14 | WY1596 |
| Lysine-limited chemostat mono-culture | 7.Line1 | 30 | Asp652 frameshift | | 14 | WY1601 |
| | | | Glu216 -> Stop | | 14 | WY1602 |
| | 7.Line2 | 30 | Pro886 -> Ser | | 11, 14, 16 | WY1603 |
| | 7.Line3 | 30 | Thr586 frameshift | | 14 | WY1604 |
| | | | Thr586 frameshift | | 14 | WY1605 |
| | 11.Line1 | 19 | Glu688 -> Stop | | 14, 16 | WY1606 |
| | | | | G(−281) ->A | – | WY1608 |
| | 11.Line2 | 19 | | A(−304) -> G | – | WY1607 |
| | | 50 | Glu793 frameshift | | 14 | WY1609 |

The online version of this article includes the following source data for Table 1:

**Source data 1.** Summary of mutations.

center and right columns bottom panels). This could allow mutants to grow faster than the ancestor during lysine limitation.

### *ecm21* mutation is partner-serving

The partner-serving phenotype of $L^-H^+$ (i.e., hypoxanthine release rate per lysine consumption; exchange ratio $r_H/c_L$) can be measured in lysine-limited chemostats. In chemostats, fresh medium containing lysine was supplied at a fixed slow flow rate (mimicking the slow lysine supply by partner), and culture overflow exited the culture vessel at the same flow rate (**Skelding et al., 2018**). After an initial lag, live and dead population densities reached a steady state (**Figure 3—figure supplement 1**), and therefore, the net growth rate must be equal to the chemostat dilution rate *dil* (flow rate/culture volume). The released hypoxanthine also reached a steady state (**Figure 3A**). The *H-L* exchange ratio can be quantified as $dil*H_{ss}/L_0$ (**Hart et al., 2019b**), where *dil* is the chemostat dilution rate, $H_{ss}$ is the steady state hypoxanthine concentration in the culture vessel, and $L_0$ is the lysine concentration in the inflow medium (which was fixed across all experiments). Note that this measure at the

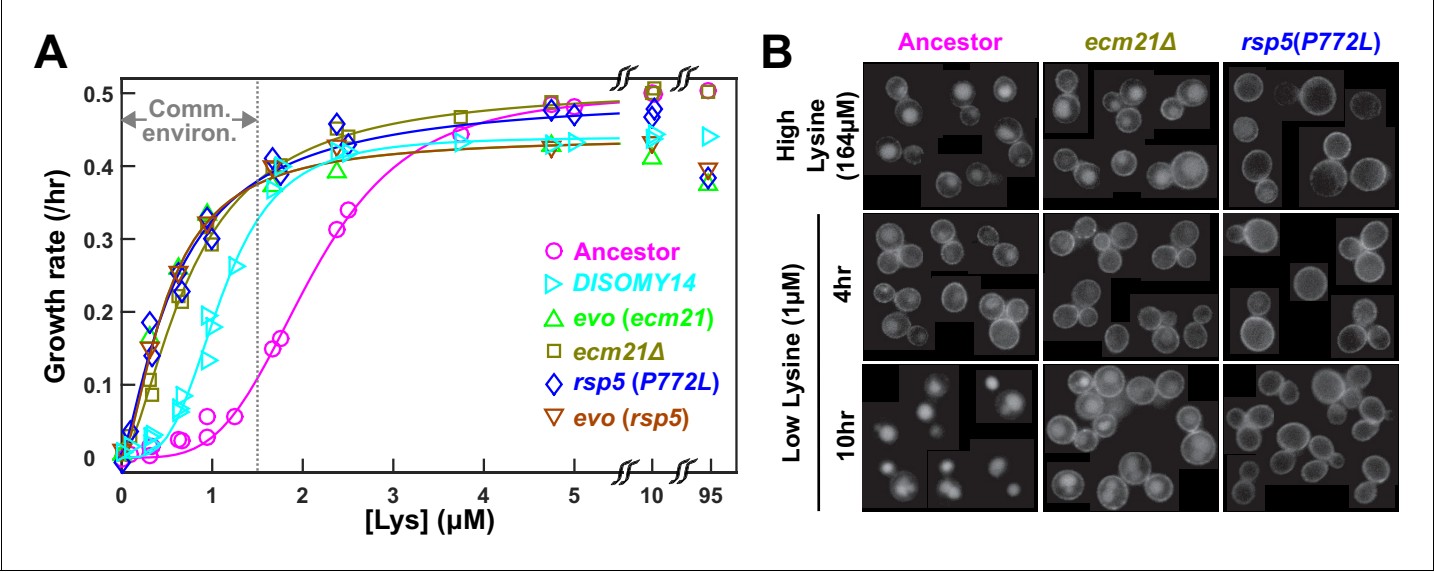

**Figure 2.** Self-serving mutations stabilize the high affinity lysine permease Lyp1 on cell membrane and improve cell growth rates at low lysine. (**A**) Recurrent mutations are self-serving. We measured growth rates of mutant and ancestral strains in minimal SD medium with various lysine concentrations, using a calibrated fluorescence microscopy assay (*Hart et al., 2019c*). Briefly, for each sample, total fluorescence intensity of image frames was tracked over time, and the maximal positive slope of ln(fluorescence intensity) against time was used as the growth rate. Evolved strains grew faster than the ancestor in community environment (the gray dotted line corresponds to the lysine level supporting a growth rate of 0.1/hr as observed in ancestral Cooperation that is Synthetic and Mutually Obligatory [CoSMO] *Hart et al., 2019a*). Measurements performed on independent days ($\geq$3 trials) were pooled and the average growth rate is plotted. Fit lines are based on Moser's equation $b(L) = b_{max}L^n/(K_L^n + L^n)$, where $b(L)$ is the cell birth rate at metabolite concentration $L$, $b_{max}$ is the maximum birth rate in excess lysine, $K_L$ is the lysine concentration at which half maximum birth rate is achieved, and $n$ is the cooperitivity cooefficient describing the sigmoidal shape of the curve (*Hart et al., 2019c*). Evolved strains are marked with *evo*; engineered or backcrossed mutants are marked with the genotype. Data for *DISOMY14* are reproduced from *Hart et al., 2019b* as a comparison. Data can be found in *Figure 2—source data 1*. (**B**) Self-serving mutations stabilize Lyp1 localization on cell membrane. We fluorescently tagged Lyp1 with GFP (green fluorescent protein) in ancestor (WY1620), *ecm21Δ* (WY2355), and *rsp5(P772L)* (WY2356) to observe Lyp1 localization. We imaged each strain in a high lysine concentration (164 μM) as well as after 4 and 10 hr incubation in low lysine (1 μM). Note that low lysine was not consumed during incubation (*Hart et al., 2019c*). During prolonged lysine limitation, Lyp1 was stabilized to cell membrane in both mutants compared to the ancestor. Images contain samples from several fields of view so that more cells can be visualized.

The online version of this article includes the following source data and figure supplement(s) for figure 2:

**Source data 1.** Growth parameters of ancestral and mutant strains.
**Figure supplement 1.** Functional domains and positions of mutations in Ecm21 and Rsp5 proteins.
**Figure supplement 2.** *ecm21Δ* rapidly outcompetes ancestor in lysine-limited chemostats.
**Figure supplement 2—source data 1.** Population dynamics of strain competition.

population level ($dil*H_{ss}/L_0$) is mathematically identical to an alternative measure at the individual level (hypoxanthine release rate per cell/lysine consumption amount per cell or $r_H/c_L$) (*Hart et al., 2019b*).

Compared to the ancestor, *ecm21Δ* but not *DISOMY14* (*Hart et al., 2019b*) or *rsp5(P772L)* exhibited increased H-L exchange ratio. Specifically, at the same dilution rate (corresponding to 6 hr doubling), the steady state hypoxanthine concentration was the highest in *ecm21Δ*, and lower in the ancestor, *DISOMY14* (*Hart et al., 2019b*), and *rsp5(P772L)* (*Figure 3A*). Although exchange ratio depends on growth rate ( = *dil*), exchange ratios of *ecm21Δ* consistently outperformed those of the ancestor across doubling times typically found in CoSMO (*Figure 3B*). Thus, compared to the ancestor, *ecm21Δ* has a higher hypoxanthine release rate per lysine consumption. This can be interpreted as improved metabolic efficiency in the sense of turning a fixed amount of lysine into a higher hypoxanthine release rate.

To test whether *ecm21Δ* can promote partner growth rate, we quantified the steady state growth rate of the $H^-L^+$ partner when cocultured with either ancestor or *ecm21Δ $L^-H^+$* in CoSMO communities. After an initial lag, CoSMO reached a steady state growth rate (*Hart et al., 2019b*; constant slopes in *Figure 4A*). The same steady state growth rate was also achieved by the two cooperating

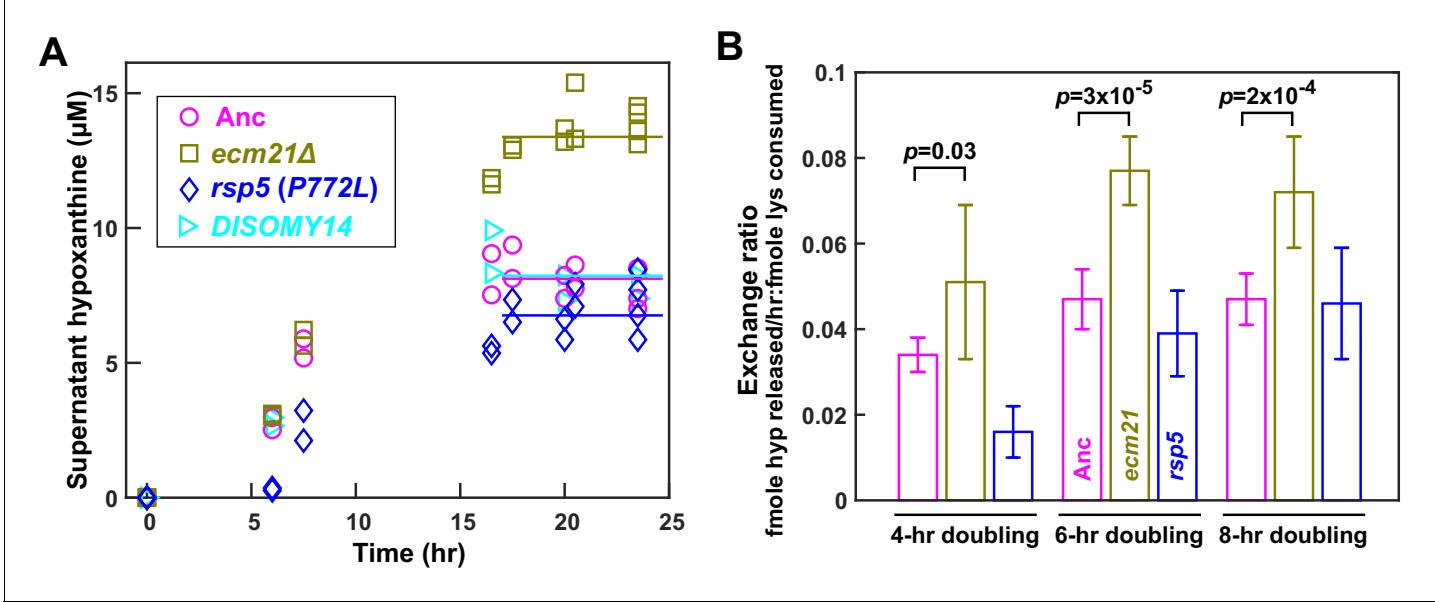

**Figure 3.** *ecm21Δ* improves hypoxanthine release rate per lysine consumption. (**A**) Hypoxanthine accumulates to a higher level in *ecm21Δ* chemostats than in ancestor chemostats. We cultured individual strains in lysine-limited chemostats (20 µM input lysine) at 6 hr doubling time (similar to Cooperation that is Synthetic and Mutually Obligatory [CoSMO] doubling time). Periodically, we quantified live and dead cell densities using flow cytometry (*Figure 3—figure supplement 1*), and hypoxanthine concentration in filtered supernatant using a yield-based bioassay (*Hart et al., 2019a*). The steady state hypoxanthine concentration created by the ancestor (WY1335) was lower than e*cm21Δ* (WY2226), and slightly higher than *rsp5(P772L)* (WY2475). *DISOMY14* (WY2349) was indistinguishable from the ancestor, similar to our previous report (*Hart et al., 2019b*). (**B**) *ecm21Δ* has a higher hypoxanthine-lysine exchange ratio than the ancestor. Cells were cultured in lysine-limited chemostats that spanned the range of CoSMO environments. In all tested doubling times, the exchange ratios of *ecm21Δ* were significantly higher than those of the ancestor. The exchange ratios of *rsp5(P772L)* were similar to or lower than those of the ancestor. Mean and two standard deviations from four to five experiments are plotted. *p*-values are from two-tailed *t*-test assuming either unequal variance (4 hr doubling) or equal variance (6 and 8 hr doublings; verified by *F*-test). Data and *p*-value calculations can be found in *Figure 3—source data 1*.

The online version of this article includes the following source data and figure supplement(s) for figure 3:

**Source data 1.** Dynamics and exchange ratios measured in chemostats.

**Figure supplement 1.** Population dynamics in chemostats.

strains (*Hart et al., 2019b*). Compared to the ancestor, *ecm21Δ* indeed sped up the steady state growth rate of CoSMO and of partner $H^-L^+$ (*Figure 4B*). Thus, *ecm21Δ* is partner-serving.

The partner-serving phenotype of *ecm21Δ* can be explained by the increased hypoxanthine release rate per lysine consumption, rather than the evolution of any new metabolic interactions. Specifically, the growth rate of partner $H^-L^+$ (and of community) is approximately the geometric mean of the two strains' exchange ratios, or $\sqrt{\frac{r_H}{c_L}\frac{r_L}{c_H}}$ (*Hart et al., 2019a*; *Hart et al., 2019b*). Here, the ancestral partner's exchange ratio ($\frac{r_L}{c_H}$) is fixed, while the exchange ratio of $L^-H^+$ ($\frac{r_H}{c_L}$) is ~1.6-fold increased in *ecm21Δ* compared to the ancestor (at doubling times of 6–8 hr; *Figure 3B*). Thus, *ecm21Δ* is predicted to increase partner growth rate by $\sqrt{1.6} - 1 = 26\%$ (95% confidence interval: 12–38%; *Figure 3—source data 1*). In experiments, *ecm21Δ* increased partner growth rate by ~21% (*Figure 4B*; *Figure 4—source data 1*).

In conclusion, when $L^-H^+$ evolved in nascent mutualistic communities and in chemostat monocultures in a well-mixed environment, win-win *ecm21* mutations repeatedly arose (*Table 1*). Thus, pleiotropic win-win mutations can emerge in the absence of any prior history of cooperation, and in environments unfavorable for cooperation.

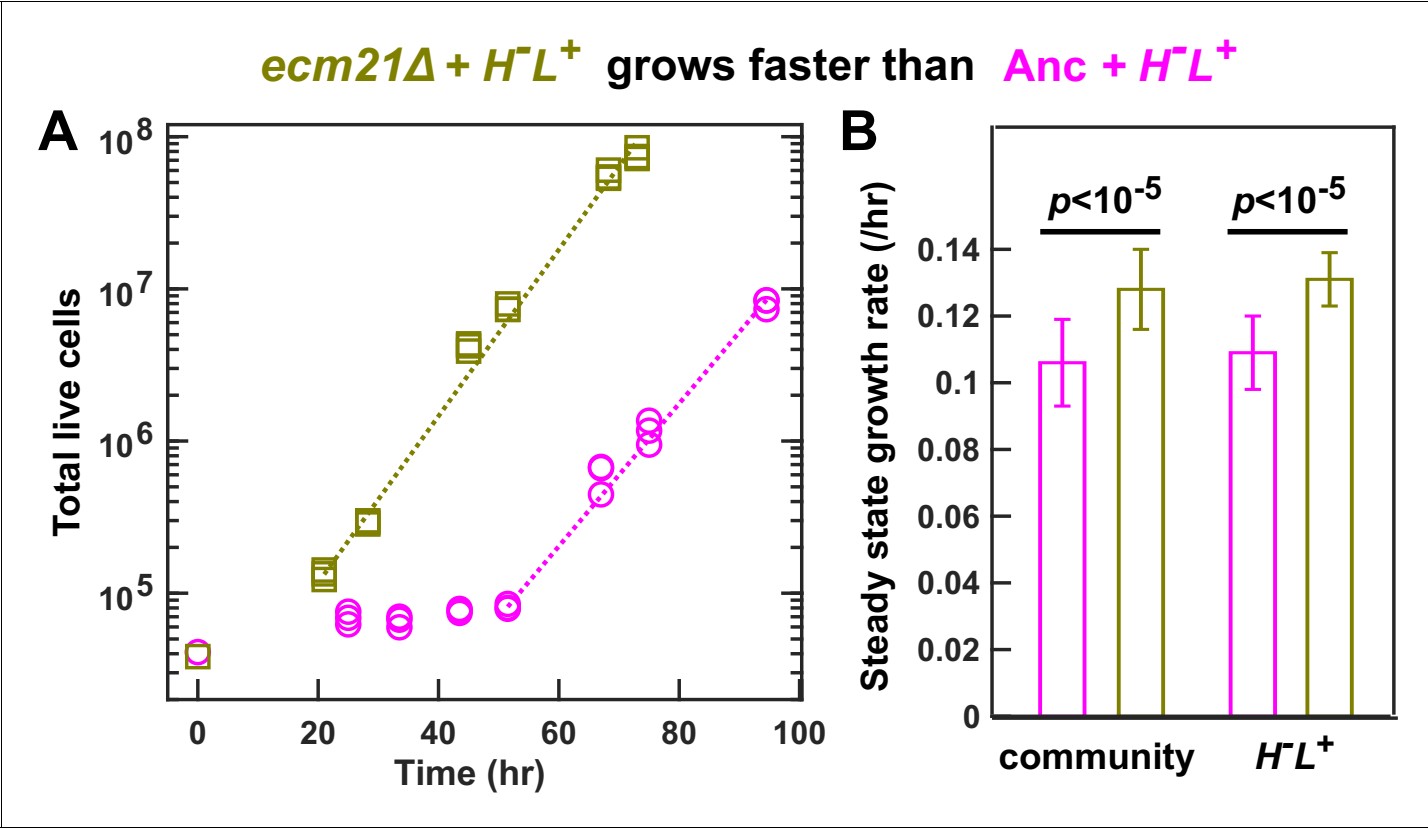

**Figure 4.** *ecm21Δ* increases the growth rate of community and of partner. To prevent rapid evolution, we grew CoSMO containing ancestral *H⁻L⁺* and ancestral or *ecm21Δ L⁻H⁺* in a spatially structured environment on agarose pads, and periodically measured the absolute abundance of the two strains using flow cytometry (*Hart et al., 2019a*). (**A**) Growth dynamics. After an initial lag, CoSMO achieved a steady state growth rate (slope of dotted line). (**B**) *ecm21Δ* increases the growth rate of CoSMO and of partner. Steady state growth rates of the entire community (left) and of partner *H⁻L⁺* (right) were measured (n ≥ 6), and the average and two standard deviations are plotted. *p*-values are from two-tailed *t*-test with equal variance (verified by *F*-test). The full data set and outcomes of statistical tests can be found in *Figure 4—source data 1*.

The online version of this article includes the following source data for figure 4:

**Source data 1.** Community growth dynamics.

## Discussion

### The evolution of win-win mutations

Here, we have demonstrated that pleiotropic win-win mutations can rapidly arise. As expected, all evolved *L⁻H⁺* clones displayed self-serving phenotypes, achieving a higher growth rate than the ancestor in low lysine presumably by stabilizing the lysine permease Lyp1 on cell membrane (*Figure 2*; *Hart et al., 2019b*). Surprisingly, *ecm21* mutants also displayed partner-serving phenotypes, promoting the steady state growth rate of partner *H⁻L⁺* and of community (*Figure 4*) via increasing the hypoxanthine release rate per lysine consumption (*Figure 3*).

The partner-serving phenotype of *L⁻H⁺* emerged as a side effect of adaptation to lysine limitation instead of adaptation to a cooperative partner. We reached this conclusion because *ecm21* mutations were also observed in *L⁻H⁺* evolving as monocultures in lysine-limited chemostats (*Table 1*). Being self-serving does not automatically lead to a partner-serving phenotype. For example, in the *DISOMY14* mutant, duplication of the lysine permease *LYP1* improved mutant's affinity for lysine (*Figure 2*) without improving hypoxanthine release rate per lysine consumption (*Figure 3A*) or partner's growth rate (*Hart et al., 2019b*).

How might *ecm21* mutants achieve higher hypoxanthine release rate per lysine consumption? One possibility is that purine overproduction is increased in *ecm21* mutants, leading to a steeper concentration gradient across the cell membrane. A different, and not mutually exclusive, possibility

is that in *ecm21* mutants, purine permeases are stabilized much like the lysine permease, which in turn leads to increased membrane permeability. Future work will reveal the molecular mechanisms of this increased exchange ratio.

The win-win effect of *ecm21* is with respect to the ancestor. *ecm21* may disappear from a population due to competition with fitter mutants. If *ecm21* is fixed in $L^-H^+$, then a new state of faster community growth (*Figure 4B*) will be established. An interesting future direction would be to investigate whether during long-term evolution of CoSMO, other win-win mutations can occur in the *ecm21* background or in backgrounds that can outcompete *ecm21*, or in the partner strain.

## How might nascent cooperation be stabilized?

A spatially structured environment is known to facilitate the origin and maintenance of cooperation (*Sachs et al., 2004*; *Chao and Levin, 1981*; *Momeni et al., 2013a*; *Harcombe, 2010*; *Nowak, 2006*; *Pande et al., 2016*; *Harcombe et al., 2018*). Here, we discuss three additional mechanisms that can stabilize nascent cooperation without partner choice capability, even when the environment is well mixed.

First, nascent cooperation can sometimes be stabilized by physiological responses to a new environment. For example, the mutualism between two metabolically complementary *Escherichia coli* strains was enhanced when one strain over-released metabolites after encountering the partner (*Hosoda et al., 2011*). Interestingly, $L^-H^+$ cells increased its hypoxanthine release rate in the presence of low lysine (which mimics the presence of cooperative partners) compared to in the absence of lysine (which mimics the absence of cooperative partners, although without lysine consumption, an exchange ratio cannot be calculated; *Hart et al., 2019a*).

Second, nascent cooperation can be stabilized by self-benefiting changes that, through promoting self-fitness, *indirectly* promote partner's fitness. Consider a mutant with improved affinity for lysine but no alterations in the metabolite exchange ratio (e.g., *DISOMY14 Hart et al., 2019b*). By growing better in low lysine, this mutant will improve its own survival which in turn helps the whole community (and thus the partner) to survive the initial stage of low cell density. Indeed, all evolved $L^-H^+$ clones tested so far improved community (and partner) survival in the sense that all mutants reduced the minimal total cell density required for the community to grow to saturation (*Waite and Shou, 2012*; *Shou et al., 2007*). Unlike *ecm21*, some of these mutations (e.g., *DISOMY14*) are not directly partner-serving and would not improve partner's steady state growth rate (*Hart et al., 2019b*).

Third, nascent cooperation can be stabilized by pleiotropic win-win mutations which *directly* promote self-fitness (by increasing competitiveness against non-mutants) and *directly* promote partner fitness (by increasing benefit supply rate per intake benefit). In this study, win-win mutations in *ecm21* rapidly evolved in a well-mixed environment, even in the absence of cooperative partner or any evolutionary history of cooperation.

## Pleiotropy and cooperation

Pleiotropic linkage between a self-serving trait and a partner-serving trait arises when both traits are controlled by the same gene (co-regulated) (*Dos Santos et al., 2018*). For example, the quorum sensing network of *Pseudomonas aeruginosa* ties together a cell's ability to make 'public goods' (such as extracellular proteases that provide a benefit to the local population) with the cell's ability to make 'private goods' (such as intracellular enzymes involved in metabolism) (*Dandekar et al., 2012*). Consequently, *LasR* mutants that 'cheat' by not secreting protease also fail to metabolize adenosine for themselves (*Dandekar et al., 2012*).

Pleiotropy might be common, given that gene networks display 'small world' connectivity (*Boone et al., 2007*) and that a protein generally interacts with many other proteins. Indeed, pleiotropic linkage between self-serving and partner-serving traits has been observed in several natural cooperative systems (*Foster et al., 2004*; *Dandekar et al., 2012*; *Oslizlo et al., 2014*; *Harrison and Buckling, 2009*; *Sathe et al., 2019*) and is thought to be important for cooperation (*Foster et al., 2004*; *Dandekar et al., 2012*; *Oslizlo et al., 2014*; *Harrison and Buckling, 2009*; *Sathe et al., 2019*; *Frénoy et al., 2013*; *Mitri and Foster, 2016*; *Chisholm et al., 2018*; *Queller, 2019*). However, such linkage can be broken during evolution (*Dos Santos et al., 2018*; *Gurney et al., 2020*). In our evolution experiments, win-win *ecm21* mutations repeatedly rose to be readily detectable in

independent lines (*Table 1*) and promoted both community growth rate (*Figure 4*) and community survival at low cell densities (*Waite and Shou, 2012*). Future work will reveal the evolutionary persistence of win-win mutations and their phenotypes.

In known examples of pleiotropic linkages between self-serving and partner-serving traits, cooperation has a long evolutionary history and is intra-population. Our work demonstrates that pleiotropy can give rise to win-win mutations that promote nascent, mutualistic cooperation. Interestingly, win-win mutation(s) have also been identified in a different engineered yeast cooperative community where two strains exchange leucine and tryptophane (*Müller et al., 2014*; Andrew Murray, personal communications). As another example, in a synthetic mutualistic community between un-engineered *E. coli* and un-engineered $N_2$-fixing *Rhodopseudomonas palustris*, a mutation in *E. coli* that improves the uptake of partner-supplied nutrients improved community growth rate (and final yield), suggesting that this mutation may also be win-win (*Fritts et al., 2020*). Overall, these observations in synthetic communities raise the possibility that pre-existing pleiotropy may have stabilized nascent cooperation in natural communities. Future work, including unbiased screens of many mutations in synthetic cooperative communities of diverse organisms, will reveal how pleiotropy might impact nascent cooperation.

## Materials and methods

### Strains
Our nomenclature of yeast genes, proteins, and mutations follows literature convention. For example, the wild-type *ECM21* gene encodes the Ecm21 protein; *ecm21* represents a reduction-of-function or loss-of-function mutation. Our *S. cerevisiae* strains are of the RM11-1a background. Both $L^-H^+$ (WY1335) and $H^-L^+$ (WY1340) are of the same mating type (*MATa*) and harbor the *ste3Δ* mutation to prevent mating between the two strains (*Table 1—source data 1*). All evolved or engineered strains used in this article are summarized in *Table 1* and *Supplementary file 1*.

Growth medium and strain culturing have been previously discussed (*Hart et al., 2019b*).

### Experimental evolution
CoSMO evolution has been described in detail in *Hart et al., 2019b*. Briefly, exponentially growing $L^-H^+$ (WY1335) and $H^-L^+$ (WY1340) were washed free of supplements, counted using a Coulter counter, and mixed at 1000:1 (Line A), 1:1 (Line B), or 1:1000 (Line C) at a total density of $5 \times 10^5$/ml. Three 3 ml community replicates (replicates 1, 2, and 3) per initial ratio were initiated, thus constituting nine independent lines. Since the evolutionary outcomes of the nine lines were similar, they could be treated as a single group. Communities were grown at 30°C in glass tubes on a rotator to ensure well mixing. Community turbidity was tracked by measuring the optical density ($OD_{600}$) in a spectrophotometer once to twice every day. In this study, 1 OD was found to be $2–4×10^7$ cells/ml. We diluted communities periodically to maintain OD at below 0.5 to avoid additional selections due to limitations of nutrients other than adenine or lysine. The fold-dilution was controlled to within 10- to 20-folds to minimize introducing severe population bottlenecks. Coculture generation was calculated from accumulative population density by multiplying OD with total fold-dilutions. Samples were periodically frozen down at −80°C. To isolate clones, a sample of frozen community was plated on rich medium YPD and clones from the two strains were distinguished by their fluorescence colors or drug resistance markers.

For chemostat evolution of $L^-H^+$, device fabrication and setup are described in detail in *Green et al., 2020*. Briefly, the device allowed the evolution of six independent cultures, each at an independent doubling time. To inoculate each chemostat vessel, ancestral $L^-H^+$ (WY1335) was grown to exponential phase in SD supplemented with 164 μM lysine. The cultures were washed with SD and diluted to $OD_{600}$ of 0.1 (~$7×10^6$/ml) in SD; 20 ml of diluted culture was added to each vessel through the sampling needle, followed by 5 ml SD to rinse the needle of excess cells. Of six total chemostat vessels, each containing ~43 ml running volume, three were set to operate at a target doubling time of 7 hr (flow rate ~4.25 ml/hr), and three were set to an 11 hr target doubling time (flow rate ~2.72 ml/hr). With 21 μM lysine in the reservoir, the target steady state cell density was $7 \times 10^6$/ml. In reality, live cell densities varied between $4 \times 10^6$/ml and $1.2 \times 10^7$/ml. Samples were periodically taken through a sterile syringe needle. The nutrient reservoir was refilled when necessary

by injecting media through a sterile 0.2 µm filter through a 60 ml syringe. We did not use any sterile filtered air and were able to run the experiment without contamination for 500 hr. Some reservoirs (and thus vessels) became contaminated after 500 hr.

Whole-genome sequencing of evolved clones and data analysis were described in detail in *Hart et al., 2019b*.

## Quantification methods

Microscopy quantification of $L^-H^+$ growth rates at various lysine concentrations was described in *Hart et al., 2019a*; *Hart et al., 2019c*. Briefly, cells were diluted into flat-bottom microtiter plates to low densities to minimize metabolite depletion during measurements. Microtiter plates were imaged periodically (every 0.5–2 hr) under a 10x objective in a temperature-controlled Nikon Eclipse TE-2000U inverted fluorescence microscope. Time-lapse images were analyzed using an ImageJ plugin Bioact (*Hart et al., 2019c*). We normalized total fluorescence intensity against that at time zero, calculated the slope of ln(normalized total fluorescence intensity) over three to four consecutive time points, and chose the maximal value as the growth rate corresponding to the input lysine concentration. For validation of this method, see *Hart et al., 2019c*.

Short-term chemostat culturing of $L^-H^+$ for measuring exchange ratio was described in *Hart et al., 2019a*; *Hart et al., 2019c*. Briefly, because $L^-H^+$ rapidly evolved in lysine-limited chemostat, we took special care to ensure the rapid attainment of steady state so that an experiment is kept within 24 hr. We set the pump flow rate to achieve the desired doubling time $T$ (19 ml culture volume*ln(2)/$T$). Periodically, we sampled chemostats to measure live and dead cell densities and the concentration of released hypoxanthine.

Cell density measurement via flow cytometry was described in *Hart et al., 2019a*. Briefly, we mixed into each sample a fixed volume of fluorescent bead stock whose density was determined using a hemocytometer or Coulter counter. From the ratio between fluorescent cells or non-fluorescent cells to beads, we can calculate live cell density and dead cell density, respectively.

Chemical concentration measurement was performed via a yield-based bioassay (*Hart et al., 2019a*). Briefly, the hypoxanthine concentration in an unknown sample was inferred from a standard curve where the final turbidities of an *ade*-tester strain increased linearly with increasing concentrations of input hypoxanthine.

Quantification of CoSMO growth rate was described in *Hart et al., 2019a*. Briefly, we used the 'spot' setting where a 15 µl drop of CoSMO community (1:1 strain ratio; ~$4 \times 10^4$ total cells/patch) was placed in a 4 mm inoculum radius in the center of a 1/6 Petri-dish agarose sector. During periodic sampling, we cut out the agarose patch containing cells, submerged it in water, vortexed for a few seconds, and discarded agarose. We then subjected the cell suspension to flow cytometry.

## Imaging of GFP localization

Cells were grown to exponential phase in SD plus 164 µM lysine. A sample was washed with and resuspended in SD. Cells were diluted into wells of a Nunc 96-well Optical Bottom Plate (Fisher Scientific, 165305) containing 300 µl SD supplemented with 164 µM or 1 µM lysine. Images were acquired under a 40× oil immersion objective in a Nikon Eclipse TE2000-U inverted fluorescence microscope equipped with a temperature-controlled chamber set at 300˚C. GFP was imaged using an ET-EYFP filter cube (Exciter: ET500/20x, Emitter: ET535/30 m, Dichroic: T515LP). Identical exposure times (500 ms) were used for both evolved and ancestral cells.

## Introducing mutations into the essential gene *RSP5*

Since *RSP5* is an essential gene, the method of deleting the gene with a drug resistance marker and then replacing the marker with a mutant gene cannot be applied. We therefore modified a two-step strategy (*Toulmay and Schneiter, 2006*) to introduce a point mutation found in an evolved clone into the ancestral $L^-H^+$ strain. First, a loxP-kanMX-loxP drug resistance cassette was introduced into ~300 bp after the stop codon of the mutant *rsp5* to avoid accidentally disrupting the remaining function in *rsp5*. Second, a region spanning from ~250 bp upstream of the point mutation [C (2315)→T] to immediately after the loxP-kanMX-loxP drug resistance cassette was PCR-amplified. The PCR fragment was transformed into a wild-type strain lacking *kanMX*. G418-resistant colonies were selected and PCR-verified for correct integration (11 out of 11 correct). The homologous

region during transformation is large, and thus recombination can occur in such a way that the transformant got the *KanMX* marker but not the mutation. We therefore Sanger-sequenced the region, found that 1 out of 11 had the correct mutation, and proceeded with that strain.

## Acknowledgements

We thank Aric Capel for the chemostat evolution experiment data, and Jose Pineda for an earlier collaboration that eventually led to this discovery. We also thank members of the Shou lab (Li Xie, David Skelding, Alex Yuan, Sonal) for discussions.

## Additional information

### Funding

| Funder | Grant reference number | Author |
| --- | --- | --- |
| National Institutes of Health | DP2 OD006498-01 | Samuel Frederick Mock Hart<br>Chi-Chun Chen<br>Wenying Shou |
| National Institutes of Health | R01GM124128 | Samuel Frederick Mock Hart<br>Wenying Shou |
| W.M. Keck Foundation | Distinguished Young Scholars | Chi-Chun Chen<br>Wenying Shou |
| Royal Society | Wolfson Fellowship | Wenying Shou |

The funders had no role in study design, data collection and interpretation, or the decision to submit the work for publication.

### Author contributions

Samuel Frederick Mock Hart, Resources, Data curation, Formal analysis, Validation, Investigation, Visualization, Methodology, Writing - review and editing; Chi-Chun Chen, Conceptualization, Formal analysis, Investigation, Visualization, Methodology, Writing - original draft, Writing - review and editing; Wenying Shou, Conceptualization, Funding acquisition, Validation, Visualization, Writing - original draft, Project administration, Writing - review and editing

### Author ORCIDs

Samuel Frederick Mock Hart (iD) https://orcid.org/0000-0002-5068-2199
Wenying Shou (iD) https://orcid.org/0000-0001-5693-381X

### Decision letter and Author response

Decision letter https://doi.org/10.7554/eLife.57838.sa1
Author response https://doi.org/10.7554/eLife.57838.sa2

## Additional files

### Supplementary files

- Supplementary file 1. List of strains.
- Transparent reporting form

### Data availability

All data in this study are included in the manuscript and supporting files. Source data files have been provided for Figures 2-4.

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
