## [Decision Letter]

**Acceptance summary:**

We appreciate how this study demonstrates that so-called pleiotropic mutations that generate a beneficial effect for the mutated individual as well as for other individuals with which the mutant cooperates, can promote the cooperative behavior. Specifically, the authors use an engineered mutualism where two types of yeast cells exchange two essential metabolites. A mutation that increases the uptake of one of these metabolites in one partner also causes increased secretion of the other metabolite, thereby also benefitting the partner's fitness. The study shows that in such (engineered) mutualistic systems, mutations that are selected because they yield a fitness benefit for the mutated partner, can also serve the non-mutated partner and thus benefit the cooperative system as a whole.

**Decision letter after peer review:**

Thank you for submitting your article "Pleiotropic win-win mutations can rapidly evolve in a nascent cooperative community despite unfavourable conditions" for consideration by *eLife*. Your article has been reviewed by three peer reviewers, one of whom is a member of our Board of Reviewing Editors, and the evaluation has been overseen by Diethard Tautz as the Senior Editor. The reviewers have opted to remain anonymous.

The reviewers have discussed the reviews with one another and the Reviewing Editor has drafted this decision to help you prepare a revised submission.

All reviewers agree that this is an interesting piece of work that shows how a mutation that is selected because it imparts a benefit, can also impart a benefit to non-mutated cells with which the mutant exchanges metabolites and/or predispose the mutant to engage in a mutualistic partnership.

However, all reviewers also raised a few salient questions, in particular about 1) the underlying molecular mechanism and 2) the exact nature and ecological and evolutionary novelty and relevance of the observation.

As far as we understand, the *ecm21* mutation increases the uptake rate of lysine because the *LYP1* permease becomes more highly expressed, thereby giving the mutants a (temporary?) fitness benefit (they are fitter than cells not carrying the *ecm21* mutation), but as far as we can tell, the yield coefficients (i.e. the efficiency of metabolism) do not change, and we would not be surprised if the *ecm21* mutants will in fact have a lower fitness (division rate) once they outcompeted all non-mutant cells because overexpression of the *LYP1* lysine permease. This would also mean that at this point, the secretion of hypoxanthine may drop back to what it was in cultures with cells not carrying *ecm21*, or perhaps even lower.

Firstly, we would ask the authors to better explain in plain wording the phenotypic consequences of the *ecm21* mutation. What does this mutation do exactly, and how does it increase fitness of the mutant? And, more importantly, how does that explain the benefit for the other genotype? What is the fitness effect in monocultures and mixed cultures over time? Is there only a positive fitness effect for *ecm21* mutants as long as WT and *ecm21* cells are present together? Does the *ecm21* mutant become fixed over time? Is a mutation really win-win if it or its benefit disappears over time? We leave it up to you to decide whether more experiments, or data analysis is needed to provide a solid answer to these questions; but in any case, we believe it is essential to re-evaluate the evolutionary framing of the story.

Second, the observation was made in artificial communities, and we would like the authors to delve deeper into the generality of their findings. Can we think of natural scenario's where similar evolutionary mechanisms could be at play?

Third, we also believe that it would help to rephrase the title to make it more broadly accessible.

Although the above summarises what we believe to be essential points, we are also providing the original review reports below since these may be helpful to you.

*Reviewer #1:*

This Research Advance builds upon previous work by the same team where they investigate the evolution of an artificial mutualistic community consisting of 2 mutant yeast lines, one secreting lysine but needing hypoxanthine (*L^+^H^-^*), and the other line relying on the secreted lysine while secreting hypoxanthine (*L^-^H^+^*). The authors found that mutants arose in the *L^-^H^+^* line that altered the rate of lysine consumption and hypoxanthine secretion per cell. Specifically, duplication of a region of Chr14 caused duplication of the *LYP1* lysine permease gene, resulting in increased lysine uptake. Similarly, duplication of the WHI3 gene that controls cell size resulted in larger cells that secrete more hypoxanthine per cell. Interestingly, while the hypoxanthine secretion per cell was increased, the secretion rate per amount of lysine remained unchanged, implying that while the mutated lines at first seem to also bestow a benefit for the *L^+^H^-^* cells due to the increased hypoxanthine release per mutant cell, the net benefit of this mutation for the *L^+^H^-^* cells is really zero, as the total number of mutant *L^-^H^+^* cells is lower, resulting in an unchanged total hypoxanthine release.

In this new Research Advance study, the authors show that a particular mutation, *ecm21*, recurrently arises among the *L^-^H^+^* cells, which results in both increased growth in limiting lysine (self-serving, increasing the growth rate of the mutated cell), as well as increased secretion of hypoxanthine (partner-serving, increasing the growth of *L^+^H^-^* cells). The *ecm21* mutations are inactivating the ECM21 gene through premature stop codons, which seems to lead to increased display of the Lyp1 permease.

These results are a bit puzzling to me; it is very well possible that I am missing something. It would help if the interpretation of what is observed would rely more on established measures such as specific yield coefficients and the specific growth speed; and that these should be measured in cultures where WT and mutants are mixed, as well as in pure cultures.

My feeling is that if the *ecm21* mutation does not affect the efficiency of the metabolism (i.e. biomass yield coefficients for hypoxanthine or lysine are unaffected by the mutation), we are looking at a Red Queen effect (i.e.: the fitness gain is only relative to the WT, but disappears when the WT cells are not present (anymore)…)

1) It is often observed that cells growing in medium with one limiting nutrient evolve mutations that increase the uptake rate (but not the metabolic efficiency), for example by increasing the expression of (functional) permeases (see for example work by Maitreya Dunham and many others). This yields a competitive advantage against wild-type, but after the wild-type is outcompeted, the growth rate drops back to wild-type levels, since there is no change in the concentration of the limiting nutrient, nor the efficiency with which the nutrient is metabolized). In evolutionary terms, this is a Red Queen effect (or even tragedy of the commons, in case cells are wasting energy to increase the uptake rate). Is this also the case here? In other words, what are the yield coefficients for biomass and hypoxanthine per lysine consumed? What is the specific growth rate of the cells (in pure and mixed cultures)? And, how this depend on the concentration of lysine? Is there only a difference when the concentration of the limiting nutrient is kept very low and near-constant, which is perhaps not a very relevant condition for a more natural setting? If this really is a Red Queen effect (or tragedy of the commons, where the growth speed of the pure mutant culture is lower than that of a pure WT culture), I think the results need to be discussed in this framework.

2) Along the same vein as the previous point: hypoxanthine release rates are measured in chemostats at a fixed dilution rate, and thus fixed cell division rate. However, is it not also necessary to correct for cell concentration? This way, you can calculate the rate per cell, instead of per population…. Given that the ecm12 mutation (temporarily) increases the growth speed, it is possible that the mutant populations are (temporarily) more dense ? That said, if the yield coefficients are unaffected, then the steady-state density may be the same (but always difficult to really reach a steady state in such biological systems…).

*Reviewer #2:*

This study from Hart et al. explores several different kinds of mutations that occur reliably in parallel populations of their engineered obligate mutualism of a lysine-secreting hypoxanthine auxotroph, and a hypoxanthine-secreting lysine auxotroph. The study identifies several different mutations that occur in replicate populations, with or without a cooperative partner present, some of which coincidentally also provide a benefit to the cooperative partner then the two are co-cultured. This last class of mutation is termed "win-win" and the main idea of the paper is that this type of mutation can be favored in the absence or the presence of the cooperative partner.

This was a very interesting paper to read and in general I had just a few broad questions for the authors to address. I don't think any of these require new experiments.

1) The paper establishes that some self-benefiting mutations, for example, those that increase lysine uptake/affinity, are positively selected purely on the basis of their direct fitness benefit and have no effect on the partner. Others, it seems by coincidence, benefit both the mutant and the partner (i.e., the win-win mutations such as *ecm21Δ*). It was explained how *DISOMY14* could be beneficial, but I don't think I saw in the manuscript where it is explained how *ecm21* is self-benefiting. Is there a known mechanistic basis for this? If not are the authors able to speculate about this? This would really help in the Discussion section.

2) From the text presented it wasn't clear to me how generalizable these results necessarily are. This is linked in part to the first note above regarding the mechanism by which *ecm21* mutation confers benefit to self. Is there a reason to suspect on the basis of metabolic network structure that the type of win-win mutation found here is something that can occur readily in many contexts, microbial or otherwise? Or is this example more likely idiosyncratic to this particular engineered mutualism? Some more detailed commentary to this effect would I think help strengthen the paper's message.

3) Is there any difference in the relative abundance of *ecm21* mutants at the end of evolution experiments in mono-culture versus experiments in co-culture with the mutualist partner? One might naively expect so given that *ecm21* mutants receive the benefit of increased partner feedback, while other purely self-benefiting mutations do not.

*Reviewer #3:*

Here the authors extend their previous work on a synthetic mutualism published in *eLife* by exploring in thoughtful detail a mutation that benefits both the focal yeast and the mutualistic partner yeast. The conceptualization and design of the experiments in the paper are strong, but there are gaps in communication and framing that, once clarified, could significantly improve the manuscript. Those areas include detailing the rationale for the sequencing choices and better anchoring expectations and results in theory and empirical data.

1) It is not particularly surprising (to me) that mutations arose in the first place or that each mutation that was beneficial to "self" either had a positive, neutral, or negative effect on the obligate partner. Instead, in my opinion, the novelty lies in whether these newly arisen mutations increase in frequency and are stable regardless of whether they evolve in response to a partner or to an abiotic metabolite source. Unfortunately, the sequencing design doesn't allow a robust answer to that question.

2) Can the authors provide some rationale for the choices made about which clones to sequence from the nine experimental replicates? My understanding is that five clones from different generations were sequenced from each of three evolution replicates.

• A1 1000:1 initial strain ratio (two clones @ gen 24 at two @ 151 gen),

• B1 1:1 initial strain ratio (one clone @ gen 25, one clone @ gen 49, three clones @ gen 76),

• and B3 1:1 initial strain ratio (two clones @ gen 14, two clones @ gen 34, one clone @ gen 63).

This sequencing design doesn't allow insight into the relative frequency of variants at a given time point (i.e., did most yeast clones at timepoint X have an *ecm21* mutation?). In fact, in 2 of the three replicates, *ecm21* mutations that were present in the early time point, were not present at the later time points.

3) The themes and results of this new work connect closely to recent modeling work by dos Santos (do Santos, Ghoul and West, 2018). Some of the key points from that work that might be relevant:

• pleiotropy only stabilizes cooperation under very narrow circumstances and these are valid for any trait and not just cooperation;

• the directionality of stabilization can occur the other way around with cooperation stabilizing pleiotropy.

4) "CoSMO models the metabolic cooperation between certain gut microbial species (Rakoff-Nahoum, Foster and Comstock, 2016) and between legumes and rhizobia (Schubert, 1986), as well as other mutualisms (Beliaev et al., 2014; Helliwell et al., 2011; Carini et al., 2014; Rodionova et al., 2015; Zengler and Zaramela, 2018; Jiang et al., 2018)." Can the authors provide more clarification for why is CoSMO is a useful model for the mutualisms listed? Many of these examples, like legumes and rhizobia, are facultative and/or spatially segregated, not obligate, and well-mixed.

5) The title is not very specific. Describing why illustrates some of my confusion with the language used in the paper more broadly.

• "Pleiotropic win-win mutations" – most readers won't know a priori that the win-win is referring to the fitness consequences to two members of a mutualism (a cooperative community could be more than two members).

• "nascent cooperative community" – nascent could mean lots of things to readers. As an evolutionary biologist, I imagine a nascent cooperative community to be one where the cooperative alleles are not fixed. Someone else might imagine a situation where cooperation is facultative. Here it refers to a synthetic, obligate mutualism.

• "despite unfavorable conditions" – are the well-mixed conditions used here a genuinely unfavorable for the maintenance of an established, obligate mutualism? I know theory suggests that mutualisms should be harder to establish in well-mixed environments, but is there empirical or theoretical evidence to back up the idea that obligate mutualisms should breakdown in less than 100 generations in a well-mixed environment?

---

## [Author Response]

[…] All reviewers agree that this is an interesting piece of work that shows how a mutation that is selected because it imparts a benefit, can also impart a benefit to non-mutated cells with which the mutant exchanges metabolites and/or predispose the mutant to engage in a mutualistic partnership.However, all reviewers also raised a few salient questions, in particular about 1) the underlying molecular mechanism and 2) the exact nature and ecological and evolutionary novelty and relevance of the observation.As far as we understand, the ecm21 mutation increases the uptake rate of lysine because the LYP1 permease becomes more highly expressed, thereby giving the mutants a (temporary?) fitness benefit (they are fitter than cells not carrying the ecm21 mutation), but as far as we can tell, the yield coefficients (i.e. the efficiency of metabolism) do not change, and we would not be surprised if the ecm21 mutants will in fact have a lower fitness (division rate) once they outcompeted all non-mutant cells because overexpression of the LYP1 lysine permease. This would also mean that at this point, the secretion of hypoxanthine may drop back to what it was in cultures with cells not carrying ecm21, or perhaps even lower.

Point 0: In limiting lysine (Figure 2A, 0.3~2 µM lysine), a pure *ecm21* culture always grows much (several-fold) faster than a pure ancestral culture. Note that the fitness difference between a mutant and the population mean is *always* a function of genotype composition and the environment. I will make a more detailed response specific to reviewer 1.

The secretion of hypoxanthine in a pure *ecm21* culture is maintained at a higher level than a pure ancestral culture (Figure 3A). The exchange ratio (release rate/consumption amount) can be interpreted as a form of yield coefficient. We have added the following to our text:

“We call this ratio “*H*-*L* exchange ratio” (Figure 1B, purple), which can be interpreted as the yield coefficient while converting lysine consumption to hypoxanthine release.”

“Thus, compared to the ancestor, *ecm21∆* has a higher hypoxanthine release rate per lysine consumption. This can be interpreted as improved metabolic efficiency in the sense of turning a fixed amount of lysine into a higher hypoxanthine release rate.”

Firstly, we would ask the authors to better explain in plain wording the phenotypic consequences of the ecm21 mutation. What does this mutation do exactly, and how does it increase fitness of the mutant? And, more importantly, how does that explain the benefit for the other genotype?

Point 1A: The *ecm21* mutation does two things: Compared to the ancestor, it 1) improves self-fitness in terms of growth rate (Figure 2A), thus outcompeting the ancestor (see the new Figure 2—figure supplement 2), and 2) improves partner growth rate (Figure 4B).

*ecm21* improves self-fitness presumably by stabilizing the lysine permease Lyp1 (Figure 2). We showed previously that duplication of Lyp1 improves cells’ ability to compete for limited lysine (Hart et al., *eLife* 2019). Your point is fair in that we did not directly show that Lyp1 stabilization explains the self-serving phenotype of *ecm21* (e.g. demonstrating that destabilizing Lyp1 renders *ecm21* ineffective when competing with wildtype in low lysine). Such experiment is challenging because if we delete *LYP1*, *L^-^H^+^* cells will not grow, and “titrating” the Lyp1 protein level to be similar to wildtype is not trivial. Thus, we have revised our text:

The self-serving phenotype is due to an increased abundance of the high-affinity lysine permease Lyp1 on the cell membrane. A parsimonious explanation for *ecm21*’s self-serving phenotype is that during lysine limitation, the high-affinity lysine permease Lyp1 is stabilized on cell surface in the mutant.

The beneficial effect of *ecm21* on partner fitness can be quantitatively explained by *ecm21*’s increased hypoxanthine release rate per lysine consumption (see the paragraph starting with “The partner-serving phenotype of *ecm21∆* can be explained by the increased hypoxanthine release rate per lysine consumption, rather than the evolution of any new metabolic interactions.”) The question of why *ecm21* can improve hypoxanthine release rate per lysine consumption is beyond the scope of this work. We quote the relevant text from the Discussion:

“How might *ecm21* mutants achieve higher hypoxanthine release rate per lysine consumption? One possibility is that purine overproduction is increased in *ecm21* mutants, leading to a steeper concentration gradient across the cell membrane. […] Future work will reveal the molecular mechanisms of this increased exchange ratio.”

Also note that:

“Being self-serving does not automatically lead to a partner-serving phenotype. For example in the *DISOMY14* mutant, duplication of the lysine permease *LYP1* improved mutant’s affinity for lysine (Figure 2) without improving hypoxanthine release rate per lysine consumption (Figure 3A) or partner’s growth rate (Hart et al., *eLife*, 2019).”

What is the fitness effect in monocultures and mixed cultures over time? Is there only a positive fitness effect for ecm21 mutants as long as WT and ecm21 cells are present together? Does the ecm21 mutant become fixed over time?

Point 1B: With respect to the fitness effect of *ecm21* in mixed cultures over time, we have added Figure 2—figure supplement 2 to show that *ecm21* is more fit than the ancestor during lysine limitation.

Is a mutation really win-win if it or its benefit disappears over time? We leave it up to you to decide whether more experiments, or data analysis is needed to provide a solid answer to these questions; but in any case, we believe it is essential to re-evaluate the evolutionary framing of the story.

Point 1C: *ecm21* is win-win with respect to the ancestor, which is the focus of our study: Can win-win mutations arise from the ancestral background? We have added the following text to the Discussion:

“The win-win effect of *ecm21* is with respect to the ancestor. *ecm21* may disappear from a population due to competition with fitter mutants. […] An interesting future direction would be to investigate whether during long-term evolution of CoSMO, other win-win mutations can occur in the *ecm21* background or in backgrounds that can outcompete *ecm21*, or in the partner strain.”

Related to this point, we have added to the Discussion:

“Pleiotropy might be common, given that gene networks display “small world” connectivity (Boone, Bussey and Andrews, 2007) and that a protein generally interacts with many other proteins. […] Future work will reveal the evolutionary persistence of win-win mutations and their phenotypes.”

Second, the observation was made in artificial communities, and we would like the authors to delve deeper into the generality of their findings.

Point 2A: The novelty of our work lies precisely in using an artificial cooperative community lacking any history of cooperation to study a question that is not easily addressable in natural cooperative communities. I have now made this more explicit by adding the following text to Introduction:

“To date, pleiotropic linkage between a self-serving trait and a partner-serving trait has been exclusively demonstrated in systems with long evolutionary histories of cooperation. […] Specifically, we test whether pleiotropic “win-win” mutations directly benefiting self and directly benefiting partner could arise in a synthetic cooperative community growing in an environment unfavorable for cooperation.”

Can we think of natural scenario's where similar evolutionary mechanisms could be at play?

Point 2B: Our work shows that pleiotropy *can* promote cooperation. By demonstrating that this phenomenon can occur in a synthetic community, our work raises the possibility that pre-existing pleiotropy may have stabilized nascent cooperation in natural communities. Clearly, concrete statements on the generality of our observation will require future work. Interestingly, Andrew Murray lab (Harvard) has found a similar phenomenon in a different synthetic yeast community. In fact, his lab and our lab have been trying to coordinate joint paper submission since 2014. To cut a very long story short, it took us a few years to figure out the right way of measuring phenotypes (Hart et al., PLoS Biology 2019) and the right definition of partner-serving phenotype (Hart et al., *eLife* 2019). Eventually, we submitted this article by ourselves since Prof. Murray informed us that he was too bogged down with administrative responsibilities. However, with his kind permission and with my recent literature search, I have added the following to the Discussion on the generality of our finding:

“Interestingly, win-win mutation(s) have also been identified in a different engineered yeast cooperative community where two strains exchange leucine and tryptophane (Muller et al., 2014) (Andrew Murray, personal communications). […] Future work, including unbiased screens of many mutations in synthetic cooperative communities of diverse organisms, will reveal how pleiotropy might impact nascent cooperation.”

Point 2C: Finally, if history is of any value in predicting the future, then CoSMO has done quite well. I have added the following to Introduction:

“Importantly, principles learned from CoSMO have been found to operate in communities of un-engineered microbes. These include how fitness effects of interactions might affect spatial patterning and species composition in two-species communities (Momeni et al., 2013), as well as how co-operators might survive cheaters (Momeni, Waite and Shou, 2013; Waite and Shou, 2012) (see Discussions in these articles).”

Third, we also believe that it would help to rephrase the title to make it more broadly accessible.

Point 3: Thanks for the great suggestion. We have revised our title to:

“Pleiotropic mutations can rapidly evolve to directly benefit self and cooperative partner despite unfavorable conditions”

Although the above summarises what we believe to be essential points, we are also providing the original review reports below since these may be helpful to you.

We appreciate your attaching the original review reports. The originals really helped us understand reviewers’ viewpoints. Below, we provide point-by-point responses.

Reviewer #1:This Research Advance builds upon previous work by the same team where they investigate the evolution of an artificial mutualistic community consisting of 2 mutant yeast lines, one secreting lysine but needing hypoxanthine (L^+^H^-^), and the other line relying on the secreted lysine while secreting hypoxanthine (L^-^H^+^). The authors found that mutants arose in the L^-^H^+^ line that altered the rate of lysine consumption and hypoxanthine secretion per cell. Specifically, duplication of a region of Chr14 caused duplication of the LYP1 lysine permease gene, resulting in increased lysine uptake. Similarly, duplication of the WHI3 gene that controls cell size resulted in larger cells that secrete more hypoxanthine per cell. Interestingly, while the hypoxanthine secretion per cell was increased, the secretion rate per amount of lysine remained unchanged, implying that while the mutated lines at first seem to also bestow a benefit for the L^+^H^-^ cells due to the increased hypoxanthine release per mutant cell, the net benefit of this mutation for the L^+^H^-^ cells is really zero, as the total number of mutant L^-^H^+^ cells is lower, resulting in an unchanged total hypoxanthine release.In this new Research Advance study, the authors show that a particular mutation, ecm21, recurrently arises among the L^-^H^+^ cells, which results in both increased growth in limiting lysine (self-serving, increasing the growth rate of the mutated cell), as well as increased secretion of hypoxanthine (partner-serving, increasing the growth of L^+^H^-^ cells). The ecm21 mutations are inactivating the ECM21 gene through premature stop codons, which seems to lead to increased display of the Lyp1 permease.

Your summary is mostly correct, except that partner-serving is increased secretion rate of hypoxanthine *per lysine consumption* (not merely increased secretion rate).

These results are a bit puzzling to me; it is very well possible that I am missing something. It would help if the interpretation of what is observed would rely more on established measures such as specific yield coefficients and the specific growth speed; and that these should be measured in cultures where WT and mutants are mixed, as well as in pure cultures.

Growth rates of evolved *ecm21* strains and of the ancestral strain in pure batch cultures are plotted in Figure 2A. Note that within the lysine-limited community environment (<1.5 μm lysine), all evolved strains grew much faster than the ancestor (see Points 0 and 1A). We have also added a supplementary figure showing that *ecm21* outcompeted the ancestor in a mixed chemostat culture (see Point 1B). Exchange ratios, which can be interpreted as yield coefficients, of pure *ecm21* and pure ancestral chemostat monocultures are plotted in Figure 3B, with *ecm21* displaying an increased hypoxanthine release rate per lysine consumed. We are unable to differentiate hypoxanthine released from different strains in a coculture.

My feeling is that if the ecm21 mutation does not affect the efficiency of the metabolism (i.e. biomass yield coefficients for hypoxanthine or lysine are unaffected by the mutation), we are looking at a Red Queen effect (i.e.: the fitness gain is only relative to the WT, but disappears when the WT cells are not present (anymore)…)

See Point 0 and Point 1C .

1) It is often observed that cells growing in medium with one limiting nutrient evolve mutations that increase the uptake rate (but not the metabolic efficiency), for example by increasing the expression of (functional) permeases (see for example work by Maitreya Dunham and many others). This yields a competitive advantage against wild-type, but after the wild-type is outcompeted, the growth rate drops back to wild-type levels, since there is no change in the concentration of the limiting nutrient, nor the efficiency with which the nutrient is metabolized). In evolutionary terms, this is a Red Queen effect (or even tragedy of the commons, in case cells are wasting energy to increase the uptake rate). Is this also the case here? In other words, what are the yield coefficients for biomass and hypoxanthine per lysine consumed? What is the specific growth rate of the cells (in pure and mixed cultures)? And, how this depend on the concentration of lysine? Is there only a difference when the concentration of the limiting nutrient is kept very low and near-constant, which is perhaps not a very relevant condition for a more natural setting? If this really is a Red Queen effect (or tragedy of the commons, where the growth speed of the pure mutant culture is lower than that of a pure WT culture), I think the results need to be discussed in this framework.

I now see the source of confusion. The concept of chemostat is not trivial to people who have not used it – I generally spend an entire lecture to teach students how chemostats work. Here, I will attempt to explain it without using math (the math part will come later if you are interested).

In chemostats, population growth rate is fixed by the input rate of the limiting metabolite. Suppose that the ancestral population grows at 8-hr doubling time at 1 μm lysine. If a chemostat is set to run at 8-hr doubling time, then the steady state concentration of lysine in the growth chamber will be 1 uM. Now, suppose that a mutant with a better affinity for lysine arises. The mutant can grow at 4-hr doubling time at 1 μm lysine, and 8-hr doubling time at 0.1 μm lysine. The mutant grows faster than the ancestor and displaces the ancestor, and in due course, lowers lysine concentration in the growth chamber. When the mutant becomes fixed, the steady state lysine concentration in the growth chamber will be 0.1 uM, and the mutant now doubles at the pre-set 8 hr doubling time.

Lysine limitation for *L^-^H^+^* and hypoxanthine limitation for *H^-^L^+^* are characteristic of the cooperative environment. If these metabolites were not limited due to external supplies, then the two strains would not cooperate but instead compete for shared nutrients (Momeni et al., 2013). Figure 3B shows improved exchange ratio in *ecm21* over a range of lysine-limited environments.

You mentioned rate-yield trade off. We did not quantify the amount of lysine consumed per gram biomass, because this term does not appear in our mathematical equation of partner growth rate. Note that since the improvement in partner growth rate can be quantitatively explained by the increased exchange ratio (see the paragraph starting with “The partner-serving phenotype of *ecm21∆* can be explained by the increased hypoxanthine release rate per lysine consumption”), other processes (if any) do not play an important role here.

2) Along the same vein as the previous point: hypoxanthine release rates are measured in chemostats at a fixed dilution rate, and thus fixed cell division rate. However, is it not also necessary to correct for cell concentration? This way, you can calculate the rate per cell, instead of per population…. Given that the ecm12 mutation (temporarily) increases the growth speed, it is possible that the mutant populations are (temporarily) more dense ? That said, if the yield coefficients are unaffected, then the steady-state density may be the same (but always difficult to really reach a steady state in such biological systems…).

Sorry about the confusion here. See Point 1A. The cell concentration has been taken into consideration in our calculations, although this term is cancelled out (release rate/cell/[consumption amount/cell]=release rate/consumption amount). We have added the following text:

“Note that this measure at the population level (*dil H*_∗ *ss*_*L*_0_ ) is mathematically identical to an alternative measure at the individual level (hypoxanthine release rate per cell/lysine consumption amount per cell or *r_H_*/*c_L_*) (Hart et al., *eLife* 2019).”

Reviewer #2:This study from Hart et al. explores several different kinds of mutations that occur reliably in parallel populations of their engineered obligate mutualism of a lysine-secreting hypoxanthine auxotroph, and a hypoxanthine-secreting lysine auxotroph. The study identifies several different mutations that occur in replicate populations, with or without a cooperative partner present, some of which coincidentally also provide a benefit to the cooperative partner then the two are co-cultured. This last class of mutation is termed "win-win" and the main idea of the paper is that this type of mutation can be favored in the absence or the presence of the cooperative partner.This was a very interesting paper to read and in general I had just a few broad questions for the authors to address. I don't think any of these require new experiments.1) The paper establishes that some self-benefiting mutations, for example, those that increase lysine uptake/affinity, are positively selected purely on the basis of their direct fitness benefit and have no effect on the partner. Others, it seems by coincidence, benefit both the mutant and the partner (i.e., the win-win mutations such as ecm21Δ). It was explained how DISOMY14 could be beneficial, but I don't think I saw in the manuscript where it is explained how ecm21 is self-benefiting. Is there a known mechanistic basis for this? If not are the authors able to speculate about this? This would really help in the Discussion section.

Sorry about this confusion. See Point 1A.

2) From the text presented it wasn't clear to me how generalizable these results necessarily are. This is linked in part to the first note above regarding the mechanism by which ecm21 mutation confers benefit to self. Is there a reason to suspect on the basis of metabolic network structure that the type of win-win mutation found here is something that can occur readily in many contexts, microbial or otherwise? Or is this example more likely idiosyncratic to this particular engineered mutualism? Some more detailed commentary to this effect would I think help strengthen the paper's message.

See Point 2B.

3) Is there any difference in the relative abundance of ecm21 mutants at the end of evolution experiments in mono-culture versus experiments in co-culture with the mutualist partner? One might naively expect so given that ecm21 mutants receive the benefit of increased partner feedback, while other purely self-benefiting mutations do not.

The takeover of *ecm21* is so rapid (since *ecm21* grows >4-fold faster than the ancestor) in both cases that we have not done this analysis. Note that partner feedback is absent in a well-mixed environment (i.e. ancestor and *ecm21* experience the same environment regardless of their contributions to the partner).

Reviewer #3:Here the authors extend their previous work on a synthetic mutualism published in eLife by exploring in thoughtful detail a mutation that benefits both the focal yeast and the mutualistic partner yeast. The conceptualization and design of the experiments in the paper are strong, but there are gaps in communication and framing that, once clarified, could significantly improve the manuscript. Those areas include detailing the rationale for the sequencing choices and better anchoring expectations and results in theory and empirical data.1) It is not particularly surprising (to me) that mutations arose in the first place or that each mutation that was beneficial to "self" either had a positive, neutral, or negative effect on the obligate partner.

Your statement is certainly correct. However, the literature does tend to focus on “cheater” mutations, because cheater is the major problem that any cooperative system must overcome. We reasoned that an unbiased view of what mutations, especially pleiotropic mutations, can do will provide a more holistic view about what can happen during the evolution of cooperation.

Instead, in my opinion, the novelty lies in whether these newly arisen mutations increase in frequency and are stable regardless of whether they evolve in response to a partner or to an abiotic metabolite source. Unfortunately, the sequencing design doesn't allow a robust answer to that question.

With our original goal in mind, we simply sequenced random clones from independent monoculture and coculture lines to identify recurrent mutations. We observed that *ecm21* showed up in both monoculture and coculture evolution. From the enormous (>4-fold) fitness advantage of *ecm21* over wildtype (see Figure 2A and Point 1B), *ecm21* will rapidly increase in frequency (saving for clonal interference), and that is why we repeatedly detected *ecm21* mutations when randomly sequencing clones. Also see Point 1C.

2) Can the authors provide some rationale for the choices made about which clones to sequence from the nine experimental replicates? My understanding is that five clones from different generations were sequenced from each of three evolution replicates.• A1 1000:1 initial strain ratio (two clones @ gen 24 at two @ 151 gen),• B1 1:1 initial strain ratio (one clone @ gen 25, one clone @ gen 49, three clones @ gen 76),• and B3 1:1 initial strain ratio (two clones @ gen 14, two clones @ gen 34, one clone @ gen 63).This sequencing design doesn't allow insight into the relative frequency of variants at a given time point (i.e., did most yeast clones at timepoint X have an ecm21 mutation?). In fact, in 2 of the three replicates, ecm21 mutations that were present in the early time point, were not present at the later time points.

We initially chose three starting strain ratios because we wanted to see whether it influenced evolutionary trajectories. We randomly sequenced a small number of clones from different generations to look for common mutations. In both coculture and monoculture lines, *ecm21* can be observed in at least some of the lines at later generations (>50 Gen). The number of clones sequenced is too small to draw any conclusions on evolutionary dynamics (except that *ecm21* repeatedly evolved). Also note that clonal interference can allow *ecm21* to exist with other mutations in a dynamic fashion, or even make *ecm21* disappear. Regardless, our goal here is to show that a win-win mutation can occur and can persist in at least some of the lines. See Point 1C.

3) The themes and results of this new work connect closely to recent modeling work by dos Santos (dos Santos, Ghoul and West, 2018). Some of the key points from that work that might be relevant:• pleiotropy only stabilizes cooperation under very narrow circumstances and these are valid for any trait and not just cooperation• the directionality of stabilization can occur the other way around with cooperation stabilizing pleiotropy

Thank you for pointing this out. It is a bit embarrassing – this project had such a long incubation time that I had forgotten that the last literature search was done years ago! I have done a new literature search, and added quite a few references. As described in Point 2A, I have added:

“To date, pleiotropic linkage between a self-serving trait and a partner-serving trait has been exclusively demonstrated in systems with long evolutionary histories of cooperation. […] A second possibility is that pleiotropy promotes cooperation (foster et al., 2004; Dandekar, Chugani and Greenberg, 2012; Oslizlo et al., 2014; Harrison and Buckling, 2009; Sathe et al., 2019). These two possibilities are not mutually exclusive.”

Also see Point 1C.

4) "CoSMO models the metabolic cooperation between certain gut microbial species (Rakoff-Nahoum, Foster and Comstock, 2016) and between legumes and rhizobia (Schubert, 1986), as well as other mutualisms (Beliaev et al., 2014; Helliwell et al., 2011; Carini et al., 2014; Rodionova et al., 2015; Zengler and Zaramela, 2018; Jiang et al., 2018)." Can the authors provide more clarification for why is CoSMO is a useful model for the mutualisms listed? Many of these examples, like legumes and rhizobia, are facultative and/or spatially segregated, not obligate, and well-mixed.

CoSMO is similar to these mutualisms because it involves costly metabolic exchange. CoSMO cooperation can also be made facultative if we supply lysine and hypoxanthine, just like legumerhizobia mutualism becoming facultative if nitrogen fertilizer is supplied. We used well-mixed environment to make the environment non-conducive for cooperation. We have now added:

“Similar to natural systems, in CoSMO exchanged metabolites are costly to produce (Waite and Shou, 2012; Hart et al., PLoS Biology 2019), and cooperation can transition to competition when the exchanged metabolites are externally supplied (Momeni et al., 2013).”

5) The title is not very specific. Describing why illustrates some of my confusion with the language used in the paper more broadly.• "Pleiotropic win-win mutations" – most readers won't know a priori that the win-win is referring to the fitness consequences to two members of a mutualism (a cooperative community could be more than two members).• "nascent cooperative community" -nascent could mean lots of things to readers. As an evolutionary biologist, I imagine a nascent cooperative community to be one where the cooperative alleles are not fixed. Someone else might imagine a situation where cooperation is facultative. Here it refers to a synthetic, obligate mutualism.• "despite unfavorable conditions" – are the well-mixed conditions used here a genuinely unfavorable for the maintenance of an established, obligate mutualism? I know theory suggests that mutualisms should be harder to establish in well-mixed environments, but is there empirical or theoretical evidence to back up the idea that obligate mutualisms should breakdown in less than 100 generations in a well-mixed environment?

Thanks for such detailed feedback. See Point 3.

With regard to your last point, if we add cheaters to the two CoSMO strains at 1:1:1 ratio in a well-mixed environment, 50% communities will crash (stop growing and eventually die) within ~20-30 doublings (Waite and Shou, 2012). In a spatially-structured environment, all cooperative communities survived by physically excluding cheaters (Momeni et al., 2013). Although this does not directly address your question, it does show that for mutualistic systems, a well-mixed environment is less desirable than a spatially-structured environment. Work from Will Harcombe also demonstrates the critical importance of a spatially-structured environment in the origin of costly mutualisms.